# Investigation of Microvascular Involvement Through Nailfold Capillaroscopic Examination in Children with Familial Mediterranean Fever

**DOI:** 10.3390/medicina61020264

**Published:** 2025-02-04

**Authors:** Fatih Kurt, Belkız Uyar, Muferet Erguven, Sengul Cangur

**Affiliations:** 1Department of Pediatrics, Duzce University, 81620 Duzce, Turkey; 2Department of Dermatology, Duzce University, 81620 Duzce, Turkey; belkizuyar@duzce.edu.tr; 3Department of Pediatric Rheumatology, Duzce University, 81620 Duzce, Turkey; muferete@yahoo.com; 4Department of Biostatistics and Medical Informatics, Duzce University, 81620 Duzce, Turkey; sengulcangur81@yahoo.com

**Keywords:** FMF, capillaroscopy, microhemorrhage, avascular areas

## Abstract

*Background and Objectives:* Familial Mediterranean fever (FMF) is a lifelong autoinflammatory disease characterized by episodes of fever and aseptic polyserositis. Commonly associated with vasculitis, FMF’s impact on microcirculation was investigated by examining nailfold capillaries using capillaroscopy. *Materials and Methods:* This study included 32 female and 28 male FMF patients diagnosed according to the Tel Hashomer and Yalçınkaya criteria and a control group of 20 female and 10 male age-matched cases. Demographic characteristics, medical history (abdominal pain, fever, chest pain, and joint pain), and physical examination findings of the cases were assessed. FMF gene mutations, acute-phase reactants, urine analysis, and spot urine protein/creatinine ratios were evaluated. Nailfold capillaries were examined via capillaroscopy by the same dermatology specialist. *Results:* There was no significant age or gender difference between groups. The most common symptoms in the case group were abdominal pain (81.7%) and joint pain (65%). Pathological findings in capillaroscopy, such as microhemorrhages and avascular areas, were significantly more frequent in the FMF case group (*p* < 0.001; *p* < 0.001). Physiological findings, including hairpin-shaped capillaries and shortened loops, were significantly more common in the control group (*p* = 0.001; *p* = 0.034). No significant relationships were found between kidney involvement, subclinical inflammation, presence of microhemorrhages and avascular areas in capillaroscopy, and disease duration. Additionally, no significant differences were observed in capillaroscopic findings between those with exon-10 mutations in the MEFV gene and those with non-exon-10 mutations. *Conclusions:* In conclusion, our study demonstrated secondary microvascular findings due to inflammation in FMF patients using capillaroscopy, a cost-effective and safe tool.

## 1. Introduction

Familial Mediterranean fever (FMF) is an autosomal recessive, monogenic inflammatory disease characterized by self-limiting episodes of fever and aseptic polyserositis [1]. The primary symptoms of FMF include recurrent fever attacks lasting 6–72 h, peritonitis, arthritis, pleuritis, and erysipelas-like erythema localized on the extensor surfaces of the leg, on the ankle joint, or on the dorsum of the foot [2]. The first attack usually occurs between the ages of 5 and 15, and 90% of patients are diagnosed before the age of 20. The frequency of attacks varies, and patients remain asymptomatic between episodes. FMF is caused by mutations in the MEFV gene, which codes for the pyrin protein. The MEFV gene is located on chromosome 16p13.3 and consists of 10 exons [3]. The human pyrin protein regulates cytokine secretion, cell death, and cytoskeletal signaling. The pyrin inflammasome is modulated by changes in Rho GTPase activity in host proteins. The PKN1 and PKN2 enzymes, which are associated with the RhoA protein, phosphorylate pyrin at specific positions, leading to pyrin binding with 14-3-3 proteins and remaining inactive. When RhoA becomes inactive, PKN1 and PKN2 activity decreases, phosphorylated pyrin levels drop, and pyrin is released from 14-3-3 proteins, forming an active inflammasome structure. The inappropriate activation of the pyrin inflammasome results in higher secretion of interleukin-1 (IL-1) and IL-18, causing systemic inflammation. Although there is increasing knowledge about the mechanisms that lead to pyrin inflammasome activation, it is still unknown why the disease is characterized by attacks. Some studies describe FMF as a disease characterized by unprovoked, spontaneous, or unpredictable attacks [4]. Mutations in exon 10 (M694V, M680I, V726A, and M694I) and exon 2 (E148Q) account for over 85% of FMF cases. The M694V mutation is known to produce the most severe clinical phenotype [3]. FMF is more common in Mediterranean populations, such as Turks, Arabs, Jews, and Armenians. In endemic countries, the prevalence of FMF ranges from 1:500 to 1:1000, with the highest prevalence being observed in Central Anatolia at 1:395 [5].

Many studies show a frequent association between FMF and vasculitis, especially IgA vasculitis and polyarteritis nodosa (PAN) [6]. Since capillaroscopic examination of nail folds is a noninvasive and reliable method that reveals microcirculation, it is widely used in vasculitis and rheumatic diseases, such as systemic sclerosis, dermatomyositis, mixed connective tissue disease, and Sjögren syndrome. This method has become a key tool in daily practice to assess disease progression and response to treatment [7].

It is likely that chronic inflammation, which persists throughout life in patients diagnosed with FMF, which is a prototype of autoinflammatory diseases, may affect microcirculation. Based on this hypothesis, we aimed to investigate microvascular involvement through capillaroscopic evaluation and to analyze its potential associations with various risk factors (age, gender, proteinuria, subclinical inflammation, personal and family medical history, response to colchicine treatment, treatment adherence, genetic mutations, and laboratory findings) in the presence of microvascular involvement.

## 2. Materials and Methods

This study included patients who presented to the Pediatric Rheumatology Clinic of Düzce University Research and Application Hospital between May 2023 and November 2023 and were diagnosed with FMF based on the Tel Hashomer [8] and Yalçınkaya [9] criteria. The study group consisted of 32 female and 28 male patients who were in an attack-free period. The patients did not have obesity, malnutrition, malignancy, or primary or secondary Raynaud’s phenomenon. The control group included 20 female and 10 male children of similar age to those in the study group who visited the well-child clinic.

The demographic characteristics of the cases (age, gender, etc.) and clinical symptoms (e.g., abdominal pain, fever, chest pain, joint pain, and physical examination findings) were evaluated. FMF gene mutations and laboratory tests, including an acute-phase reactant test (e.g., complete blood count (CBC), erythrocyte sedimentation rate, C-reactive protein (CRP), and serum amyloid A (SAA) levels), urine analysis, and spot urine protein/creatinine ratio assessment, were evaluated. Additionally, the nailfold capillaries of these patients were examined by a dermatologist using capillaroscopy. With the consideration that it might cause trauma in the microvascular area, the presence of nail-biting habits in the cases was evaluated. All evaluations were documented.

Nailfold capillaroscopy (NFC) was performed in accordance with previously published guidelines [10]. After waiting for 15–20 min at standard room temperature (20–22 °C), each patient was seated comfortably with their hands at heart level. The capillaroscopic evaluation of all cases was conducted by the same dermatologist, who was blinded to whether the cases belonged to the control or patient group. Although the literature indicates that the best morphological evaluation is typically obtained from the fourth and fifth fingers, NFC was performed on all fingers to account for potential early microvascular changes [10]. However, a significant part of the thumbs in both the patient and control groups was excluded from analysis due to issues such as pigmentation and trauma that impaired evaluation. Ultrasound gel was used as a coupling medium to enhance the transparency and resolution of the images. A nailfold capillaroscopic examination of all cases was recorded at 30× magnification (10 × 3) in video mode on an iPhone 13 connected to a DermLite DL5 (Dermlite, San Juan Capistrano, CA, USA) handheld dermatoscope. The nailfolds of all patients were assessed for qualitative parameters, including hairpin-shaped capillaries, capillary dilatation, avascular areas, extravasations (microhemorrhages), cross-linked capillaries, tortuous capillaries, elongated capillaries, neoangiogenesis, and shortened loops [10].

The findings of the nailfold capillaries observed through capillaroscopy were classified as follows:


**X**
**: Could not be evaluated**


X1: Could not be evaluated (pigment darkness, trauma);

X2: Nail deformity.


**A: Hairpin-shaped capillaries**


Hairpin-shaped capillaries were interpreted as normal capillaries and were described as uniform hairpin-shaped loops that were often seen arranged in a parallel row (Figure 1) [11].


**B: Avascular**
**areas**


Avascular areas could be defined as the absence of two or more successive capillaries. The avascular score was assessed using a semiquantitative scale described by Lee et al., ranging from 0 to 3:

B0—absence of avascular areas;

B1—discrete devascularization (one or two avascular areas);

B2—moderate devascularization (more than two avascular areas);

B3—extensive and confluent devascularization areas (Figure 2) [12].


**D**
**: Capillary dilatation**


0: Absent;

1: Focal enlarged capillaries or frequent apically enlarged capillaries;

2: Frequent enlarged capillaries or the presence of megacapillaries [13].


**H**
**: Extravasations (microhemorrhages**
**)**


A semiquantitative scale was used, ranging from 0 to 2.

0: Absent or clearly traumatic;

1: Focal dotty microhemorrhages;

2: Clustered (more than three lesions) microhemorrhages or frequent dotty microhemorrhages (Figure 3) [13].


**C**
**: Cross-linked capillaries**


Cross-linked capillaries were defined as intersecting branches that resembled the number 8 [11].


**T: Tortuous capillaries**


Tortuous capillaries were defined as capillaries with branches in a wavy, winding, or twisted arrangement.

0: Normal (tortuosity in less than 10% of the loops);

1: Slight (tortuosity in more than 10% of loops or focal crossed loops);

2: Marked (frequent crossed loops or the presence of bushy capillaries) [13].


**E**
**: Elongated capillaries**



**N:**
** Neoangiogenesis**


This is a new capillary formation that is seen as ramified, meandering, or bushy capillaries. In adjacencies of reduced capillary density or avascular areas, neovascularization is accompanied by the development of branching capillaries [10].

**J: Shortened loops:** nonspecific capillary changes or mild normal (Figure 4) [14,15].

DNA samples obtained from peripheral blood of FMF cases were analyzed using the next-generation sequencing (NGS) method. Exons and exon–intron junctions (±10 bp) were included in the analysis. Pathogenicity classification of the resulting data was conducted according to the ACMG Guidelines (PMID: 25741868). The reference genome used was hg19, and the ClinVar, Franklin, and Varsome databases were utilized.

Proteinuria was defined as a spot urine protein/creatinine ratio (PCR) of >0.2 mg/mg or a positive result for protein in a urine test with a dipstick [16]. In patients in which proteinuria was detected, the diagnosis was confirmed by reassessing protein levels in a 24 h urine sample within 3 m.

Subclinical inflammation was defined as plasma CRP and SAA levels and erythrocyte sedimentation rate that were elevated above the reference values without any clinical signs or symptoms associated with FMF [17].

Cases with known inflammatory diseases, Raynaud’s phenomenon, connective tissue diseases, malignancies, or malnutrition were excluded from the study.

Before inclusion in the study, the parents of the cases were informed about the study’s content, purpose, and procedures, and their written consent was obtained.

This research involving human subjects complied with all relevant national regulations and institutional policies and was conducted in accordance with the tenets of the Helsinki Declaration. This study was approved by the Duzce University Faculty of Medicine Ethics Committee (Decision no: 2023/45, Approval Date: 20 March 2023).

### 2.1. Statistical Analysis

The quantitative variables in this study are presented as the mean ± standard deviation, while the qualitative data are shown as counts and percentages. The normality assumption for quantitative variables was assessed using skewness, kurtosis coefficients, and the Shapiro–Wilk test. The assumption of homogeneity of group variances was checked with Levene’s test. An independent-sample t-test was used for intergroup comparisons of quantitative variables. Relationships between categorical variables were examined using the Pearson’s Chi-square and Fisher’s exact tests (with post-hoc Bonferroni-corrected z tests). Chi-square and binomial tests were applied for comparisons between ratios. The point biserial correlation coefficient was calculated to examine relationships between variables. Statistical evaluations were performed using SPSS version 22, and a *p*-value < 0.05 was considered statistically significant.

### 2.2. Findings

Of the 90 children included in the study, 57.8% were female, with a mean age of 12 ± 3.6 years (range 5–18). The groups were homogeneous in terms of gender and age (*p* > 0.05, Table 1). The proportion of children who bit their nails was significantly higher in the patient group (60%) compared with the control group (30%) (*p* < 0.05).

The patients’ symptoms and laboratory results are presented in Table 2. There was a significant difference in the frequency of each symptom among the patients, especially with the occurrence rates of abdominal pain (81.7%) and joint pain (65%) being significantly higher (*p* < 0.05).

The patients in our cohort had been diagnosed with FMF for an average of 6.4 ± 2.6 years (range: 1–14 years). The adherence rates to colchicine treatment among the patients showed a statistically significant difference, with a higher proportion adhering to the treatment (74.6%) compared with those not adhering (*p* < 0.05); however, there was no significant difference in response rates to colchicine treatment (*p* > 0.05, Table 3). The proportion of patients with a family history of FMF (78.3%) was significantly higher than that of those without such a history (21.7%) (*p* < 0.05). The incidence rates of proteinuria (26.7%) and subclinical inflammation (11.7%) were significantly lower than those in patients without these conditions (*p* < 0.05). Among the cases, three patients had hypertension, one had Müllerian agenesis, one had Behçet’s disease, one had Henoch–Schönlein purpura, one had nephrolithiasis, and one had a urinary double-collecting system anomaly as comorbid conditions (Table 3). Among our cases, 51.7% had MEFV exon-10 mutations (M694V, M680I, M694I, K695R, V726A, A744S, and R761H), while 48.3% had non-exon-10 mutations (E148Q, P369S, and F479L) (Table 3).

The capillaroscopic evaluation results of the groups are presented in Table 4. Among the groups, only the prevalence rates of hairpin-shaped capillaries (A), avascular areas (B1, B2, and B3), microhemorrhages (H1 and H2), and shortened loops (J) showed significant differences (*p* < 0.05, Table 4). Compared with the control group, the patient group had significantly lower rates of normal capillaries (A) and shortened capillaries (J), while the prevalence rates of mild, moderate, and large avascular areas (B1, B2, and B3) and microhemorrhages (H1 and H2) were significantly higher (*p* < 0.05). Capillary dilatation was not detected in any of our cases. No significant differences were found in other comparisons (*p* > 0.05).

In all cases (*p* = 0.581) and in the patient group (*p* = 0.481), no significant relationship was found between the presence of neoangiogenesis and age, whereas a significant negative correlation was observed in the control group (r = −0.378, *p* = 0.039).

Table 5 presents descriptive statistics for disease duration based on certain clinical characteristics and capillaroscopic findings. No significant difference was observed in disease duration with respect to the presence of renal involvement or subclinical inflammation (*p* > 0.05, Table 5). Additionally, no significant difference in disease duration was found according to the presence of B1, B2, B3, H1, H2, or J capillaroscopic findings (*p* > 0.05, Table 5).

The capillaroscopic results of patients with and without MEFV exon-10 group mutations are presented in Table 6. There was no significant difference in the prevalence of capillaroscopic findings B1, B2, B3, H1, H2, and J based on the exon-10 grouping (*p* > 0.05) (Table 6).

In Table 7, capillaroscopic findings are presented based on the response and adherence to colchicine treatment in the patient group. No statistically significant differences were observed in the prevalence rates of capillaroscopic findings B1, B2, B3, H1, H2, and J according to the treatment response and adherence (*p* > 0.05, Table 7).

In Table 8, capillaroscopic findings are presented based on the presence of comorbidities in the patient group. No statistically significant differences were found in the prevalence rates of capillaroscopic findings A, B1, B2, B3, H1, H2, and J according to the presence of comorbidities (*p* > 0.05, Table 8).

## 3. Discussion

FMF is an autosomal recessive disease characterized by self-limiting episodes of fever and serositis. The most common symptoms of FMF vary across studies. In a study by Çekin et al., abdominal pain was reported in 76%, and fever was reported in 58% of cases as the most frequent symptoms [18]. In a study by Di Ciaula et al. in Italy, fever was observed in 84%, joint pain was observed in 66%, and abdominal pain was observed in 56% of the cases [19]. In our cohort, the most common symptoms were abdominal pain, which was seen in 81.7%, and joint pain, which was seen in 65% of the cases. In a study by Çakın et al., elevated SAA levels were found in 28.5% of FMF cases [20]. Similarly, in our study, elevated SAA levels were found in 27.9% of the cases.

The frequency of MEFV gene mutation carrier status can reach as high as 1:3 to 1:10 in high-risk populations. Due to the high carrier rate, pseudodominant inheritance can be observed, particularly in communities where consanguineous marriages are common. Some studies have also suggested that FMF could be a dominant disease with low penetrance. This explains the vertical transmission of FMF within families and its clinical phenotype [21]. In our study, it was found that 78.3% of the cases had a family member diagnosed with FMF. The common occurrence of consanguineous marriages in our community and the possibility of FMF being a dominant condition with low penetrance help explain this vertical transmission.

In a study by Sakallı et al., proteinuria was detected in 28% of pediatric patients and 52% of adult patients. Although proteinuria is often associated with amyloidosis, amyloidosis was found in only 16.6% of pediatric patients with proteinuria and 69.2% of adult patients with proteinuria. The study suggested that due to ongoing subclinical inflammation, even during attack-free periods in FMF patients, proteinuria associated with amyloidosis could develop as the disease duration increases [22]. In our study, proteinuria was detected in 26.7% of the cases. However, since kidney biopsies were not performed, it was not possible to determine how much of the proteinuria was related to amyloidosis.

Vasculopathies are associated with various immune-mediated conditions, such as autoimmune diseases and immunodeficiencies. In autoinflammatory diseases, the spontaneously occurring and escalating inflammatory response results from a lack of mechanisms that limit the immune response. In this context, blood vessels are affected, leading to the development of various types of vasculitis [23]. Capillaroscopy is frequently used in rheumatological diseases such as vasculitis and systemic sclerosis, as it visualizes the microcirculation. It has become a crucial method in daily practice for assessing disease progression and response to treatment [7]. Capillaroscopy is a simple, cost-effective, noninvasive, and practical tool for evaluating microvascular structures. Due to the parallel alignment of the capillaries’ main axis with the skin surface and the accessibility of the area, nailfolds are highly suitable for the in vivo morphological assessment of skin capillaries.

The frequent coexistence of FMF and vasculitis indicates that the disease may involve not only serous membranes but also vascular pathologies. Among these, FMF is most commonly associated with IgA vasculitis. In such vasculitides, endothelial damage leads to the extravasation of erythrocytes into the extravascular space in clusters, resulting in focal and confluent microhemorrhagic areas [6]. A study by Karaca et al. reported that persistent subclinical inflammation between FMF attacks contributes to endothelial dysfunction, which is associated with increased atherothrombosis and platelet activation. This process affects the choroidal and retinal vessels, leading to an enlarged foveal avascular zone and reduced choroidal thickness [24]. It has been suggested that the capillaroscopic evaluation of FMF patients could provide insights into FMF-related vascular pathologies. The first study on the capillaroscopic evaluation of FMF patients was conducted by Dinç et al. In this study, adult FMF patients were compared with patients with systemic lupus erythematosus, systemic sclerosis, and healthy controls. It was reported that microvascular changes were detected in the FMF group [13]. Similarly, a study by Aytekin et al. on adults compared FMF patients with healthy controls using capillaroscopic evaluations. This study also reported the presence of microvascular morphological changes in FMF [25].

To our knowledge, the only study on nailfold capillaroscopic evaluation in pediatric FMF patients was conducted by Dursun et al. In this study, FMF patients in the attack phase, those in remission, and healthy controls were compared in terms of capillaroscopic findings. It was reported that patients in the attack phase showed significantly more microvascular abnormalities than those in remission and those in the control group, with a significant positive correlation between microvascular abnormalities and inflammatory processes. However, the specific capillaroscopic findings observed were not detailed [26]. In our study, avascular areas and microhemorrhagic abnormalities were found more frequently in the patient group, while normal capillary structures were observed less frequently. These findings support the presence of microvascular abnormalities in FMF patients and indicate which pathological findings may be detected. A significant limitation of our study is the lack of standardization in normal and abnormal capillaroscopic findings in pediatric cases. Capillaroscopic findings were evaluated only through nailfolds, and a more comprehensive vascular examination could explore microvascular changes in other areas as well. Studies with a larger sample size may provide more robust statistical results. Additionally, the fact that the study was conducted at a single center may limit the generalizability of the results to the broader population.

Shortened loops have been described as nonspecific capillary changes in some studies and as mild and normal findings in others [14,15]. In our study, they were detected significantly more frequently in the control group than in the patient group. This finding suggests that shortened loops may be considered a normal capillaroscopic finding.

In a study by Dursun et al., it was reported that the distribution of MEFV gene mutations and homozygosity does not contribute to capillary abnormalities [26]. In our study, when we compared patients with and without MEFV exon-10 mutations, we also found no significant differences in terms of microvascular abnormalities. Patients with exon-10 mutations are known to have a more severe clinical course. The presence of similar capillaroscopic findings in patients with non-exon-10 mutations suggests that, although attacks are less frequent in this group, the level of persistent inflammation may be similar to that in patients with exon-10 mutations. Although detailed genetic analyses of MEFV mutations in FMF patients were performed, the effects of genetic variations on microvascular changes could be further explored through larger and more detailed genetic studies.

Colchicine is the cornerstone of treatment in FMF, as it prevents attacks and amyloidosis. However, despite regular colchicine use, a significant number of patients experience a persistent inflammatory process. This persistent inflammation is asymptomatic and represents a subtle characteristic of FMF. It can lead to complications such as amyloidosis, renal failure, and infertility without overt attacks [27]. In our study, no significant differences were found in capillaroscopic findings between patients who adhered to colchicine treatment and those who did not or between those who derived full or partial benefit from the treatment. Persistent inflammation despite colchicine therapy may explain the frequent occurrence of capillaroscopic findings. During inflammatory processes, elevated levels of pro-inflammatory cytokines can trigger endothelial damage, leading to the disruption of microvascular structures [28]. Microvascular changes associated with chronic inflammation in FMF patients may not fully resolve, even after treatment. Patients undergoing colchicine treatment may not notice or may disregard recurrent mild inflammations. Environmental and familial factors can also influence persistent inflammation. This study may not provide sufficient information on the long-term effects of colchicine therapy and the complications of persistent inflammation in the long term.

## 4. Conclusions

In conclusion, our study demonstrated microvascular changes in FMF patients using capillaroscopy, an inexpensive and safe tool, and the findings supported the presence of chronic inflammation in FMF. Furthermore, to the best of our knowledge, this study is the first to describe microvascular abnormalities that can be observed in pediatric FMF patients. The results, which indicate that persistent inflammation continues in some patients despite colchicine treatment, emphasize that FMF is a subtle and chronic inflammatory disease.

## Figures and Tables

**Figure 1 medicina-61-00264-f001:**
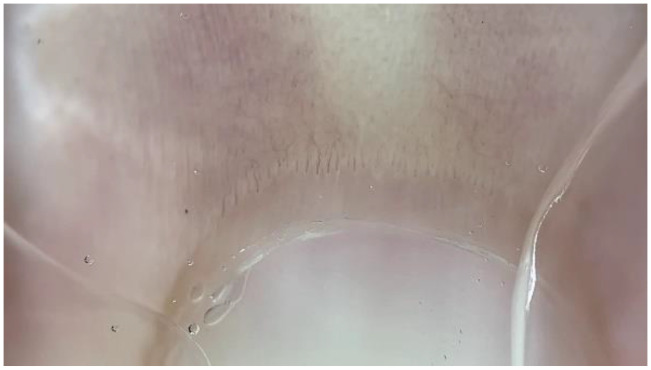
Normal nailfold capillaries of a healthy individual presenting with a regular hairpin shape (dermatoscope: DermLite DL5, polarized ×30).

**Figure 2 medicina-61-00264-f002:**
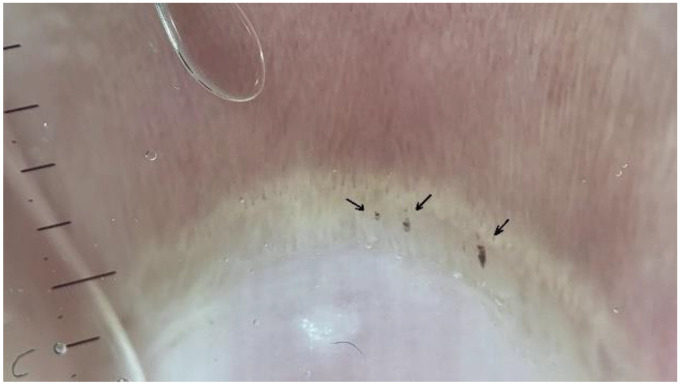
Multiple pericapillary microhemorrhages of an FMF patient (black arrows; dermatoscope: DermLite DL5, polarized ×30).

**Figure 3 medicina-61-00264-f003:**
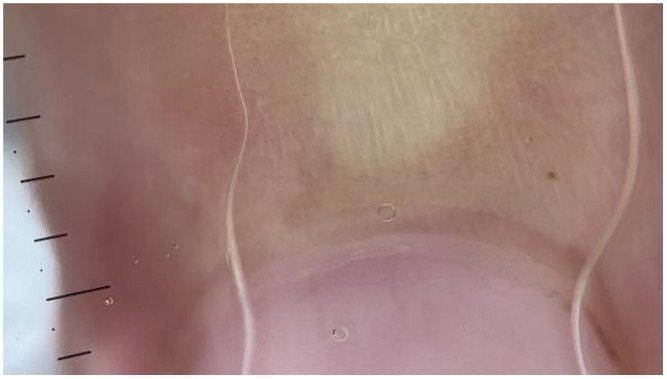
NFC from a patient with FMF showing avascular areas (dermatoscope: DermLite DL5, polarized ×30).

**Figure 4 medicina-61-00264-f004:**
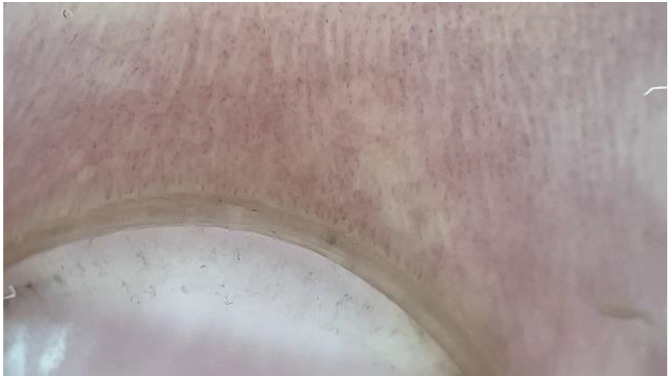
NFC from a patient with FMF showing shortened loops (dermatoscope: DermLite DL5, polarized ×30).

**Table 1 medicina-61-00264-t001:** Comparison of the sociodemographic characteristics and nail-biting habits of the cases.

		Groups						*p*
		Patient (*n* = 60)		Control (*n* = 30)		Total (*n* = 90)		
		*n*	%	*n*	%	*n*	%	
Gender	Boy	28	46.7	10	33.3	38	42.2	0.227
	Girl	32	53.3	20	66.7	52	57.8	
Nail-Biting Habit	Presence	36	60	9	30	45	50	0.007
	Absence	24	40	21	70	45	50	
Age *		11.9 ± 3.6 (5–18)		12.2 ± 3.7 (5–17)		12 ± 3.6(5–18)		0.697

* Mean ± standard deviation (minimum–maximum).

**Table 2 medicina-61-00264-t002:** Symptoms and laboratory results of the patient group.

			*n*	%	*p*
Symptoms	Fever	Yes	21	35	0.027
		No	39	65	
	Abdominal Pain	Yes	49	81.7	<0.001
		No	11	18.3	
	Joint Pain	Yes	39	65	0.027
		No	21	35	
	Chest Pain	Yes	4	6.7	<0.001
		No	56	93.3	
	Erysipelas-Like Rash	Yes	1	1.7	<0.001
		No	59	98.3	
	Myalgia	Yes	2	3.3	<0.001
		No	58	96.7	
Laboratory Results	Hemoglobin (g/dL) *		12.3 ± 1.4		
	Leukocyte Count (×10^3^/µL) *		6.9 ± 2		
	Platelet Count (×10^3^/µL) *		299 ± 67.5		
	CRP (mg/dL) *		0.6 ± 1.5		
	Erythrocyte Sedimentation Rate (mm/h) *		14.8 ± 12		
	SAA Levels (mg/L)		10.4 ± 19.7		
	High SAA Levels	Yes	12	27.9	
		No	31	72.1	

* Mean ± standard deviation; CRP: C-reactive protein; SAA: serum amyloid A; µL: microliter; mg/dL: milligrams/deciliter; mm/h: millimeters/hour.

**Table 3 medicina-61-00264-t003:** Clinical characteristics and genetic mutations of the patient group.

		*n*	%	*p*
Response to colchicine treatment	Yes	36	61	0.117
	Partial	23	39	
Adherence to colchicine treatment	Yes	44	74.6	<0.001
	No	4	6.8	
	Partial	11	18.6	
Proteinuria	Yes	16	26.7	<0.001
	No	44	73.3	
Subclinical inflammation	Yes	7	11.7	<0.001
	No	53	88.3	
Comorbidities	Yes	8	13.3	<0.001
	No	52	86.7	
Familial history	Yes	47	78.3	<0.001
	No	13	21.7	
Gene exon-10 group	Exon-10	31	51.7	0.897
	non-exon-10	29	48.3	
FMF onset age *		5.5 ± 3.2 (1–15)		
FMF diagnosis age *		6.2 ± 3.6 (1–15)		
Disease duration *		6.4 ± 2.6 (1–14)		

* Mean ± standard deviation (minimum–maximum).

**Table 4 medicina-61-00264-t004:** Comparison of capillaroscopic evaluations in the cases.

			Group						*p*
			Patient		Control		Total		
			*n*	%	*n*	%	*n*	%	
A (Hairpin-shaped capillaries)		Yes	34	56.7	29	96.7	63	70	<0.001
		No	26	43.3	1	3.3	27	30	
B (Avascular areas)	B1	Yes	47	78.3	0	0	47	52.2	<0.001
		No	13	21.7	30	100	43	47.8	
	B2	Yes	39	65	9	30	48	53.3	0.002
		No	21	35	21	70	42	46.7	
	B3	Yes	34	56.7	0	0	34	37.8	<0.001
		No	26	43.3	30	100	56	62.2	
H (Microhemorrhages)	H0	Yes	10	16.7	3	10	13	14.4	0.532
		No	50	83.3	27	90	77	85.6	
	H1	Yes	33	55	0	0	33	36.7	<0.001
		No	27	45	30	100	57	63.3	
	H2	Yes	12	20	0	0	12	13.3	0.007
		No	48	80	30	100	78	86.7	
C (Cross-linked capillaries)		Yes	4	6.7	0	0	4	4.4	0.297
		No	56	93.3	30	100	86	95.6	
T (Tortuous capillaries)	T0	Yes	2	3.3	1	3.3	3	3.3	0.999
		No	58	96.7	29	96.7	87	96.7	
	T1	Yes	0	0	1	3.3	1	1.1	0.333
		No	60	100	29	96.7	89	98.9	
E (Elongated capillaries)		Yes	4	6.7	0	0	4	4.4	0.297
		No	56	93.3	30	100	86	95.6	
N (Neoangiogenesis)		Yes	20	33.3	7	23.3	27	30	0.329
		No	40	66.7	23	76.7	63	70	
J (Shortened loops)		Yes	20	33.3	17	56.7	37	41.1	0.034
		No	40	66.7	13	43.3	53	58.9	

**Table 5 medicina-61-00264-t005:** Relationship of disease duration and clinical features with capillaroscopic findings.

		Disease Duration			*p*
		Median	Q1	Q3	
Proteinuria	Yes	6	4.5	7	0.500
	No	6	5	8	
Subclinical inflammation	Yes	6	4	9	0.735
	No	6	5	8	
Capillaroscopic findings					
B1	Yes	6	5	8	0.971
	No	6	4	7	
B2	Yes	6	5	8	0.191
	No	7	6	8	
B3	Yes	6	5	8	0.804
	No	6	4	7	
H1	Yes	6	5	7	0.449
	No	6	5	9	
H2	Yes	6	4	6.5	0.176
	No	6	5	8.5	
J	Yes	6.5	5	8	0.579
	No	6	4	8	

Q1: 1. Quartile; Q3: 3. Quartile.

**Table 6 medicina-61-00264-t006:** Relationship between genetic mutations and capillaroscopic findings in the patient group.

		Genetic Test Groups				*p*
		Exon 10		Non-Exon-10		
		*n*	%	*n*	%	
B1	Yes	23	74.2	24	82.8	0.421
	No	8	25.8	5	17.2	
B2	Yes	23	74.2	16	55.2	0.123
	No	8	25.8	13	44.8	
B3	Yes	19	61.3	15	51.7	0.455
	No	12	38.7	14	48.3	
H1	Yes	15	48.4	18	62.1	0.287
	No	16	51.6	11	37.9	
H2	Yes	8	25.8	4	13.8	0.245
	No	23	74.2	25	86.2	
J	Yes	11	35.5	9	31.0	0.715
	No	20	64.5	20	69.0	

**Table 7 medicina-61-00264-t007:** Relationship between colchicine treatment adherence, treatment response, and capillaroscopic findings in the patient group.

		Response to Treatment ^&^				Adherence to Treatment ^#^						*p*
		Yes		Partial		Yes		No		Partial		
		*n*	%	*n*	%	*n*	%	*n*	%	*n*	%	
B1	Yes	27	75	19	82.6	34	77.3	3	75	9	81.8	0.492 ^&^ 0.999 ^#^
	No	9	25	4	17.4	10	22.7	1	25	2	18.2	
B2	Yes	25	69.4	14	60.9	29	65.9	3	75	7	63.6	0.497 ^&^ 0.999 ^#^
	No	11	30.6	9	39.1	15	34.1	1	25	4	36.4	
B3	Yes	23	63.9	10	43.5	23	52.3	3	75	7	63.6	0.124 ^&^ 0.659 ^#^
	No	13	36.1	13	56.5	21	47.7	1	25	4	36.4	
H1	Yes	20	55.6	13	56.5	24	54.5	2	50	7	63.6	0.942 ^&^ 0.901 ^#^
	No	16	44.4	10	43.5	20	45.5	2	50	4	36.4	
H2	Yes	7	19.4	5	21.7	9	20.5	0	0	3	27.3	0.999 ^&^ 0.739 ^#^
	No	29	80.6	18	78.3	35	79.5	4	100	8	72.7	
J	Yes	13	36.1	7	30.4	13	29.5	1	25	6	54.5	0.653 ^&^ 0.272 ^#^
	No	23	63.9	16	69.6	31	70.5	3	75	5	45.5	

^&^: Comparison by response to treatment. ^#^: Comparison by adherence to treatment.

**Table 8 medicina-61-00264-t008:** Comparison of capillaroscopic findings in cases with and without comorbidities.

		Comorbidity				*p*
		Yes		No		
		*n*	%	*n*	%	
A	Yes	6	75	28	53.8	0.446
	No	2	25	24	46.2	
B1	Yes	6	75	41	78.8	0.999
	No	2	25	11	21.2	
B2	Yes	7	87.5	32	61.5	0.241
	No	1	12.5	20	38.5	
B3	Yes	5	62.5	29	55.8	0.999
	No	3	37.5	23	44.2	
H1	Yes	3	37.5	30	57.7	0.448
	No	5	62.5	22	42.3	
H2	Yes	1	12.5	11	21.2	0.999
	No	7	87.5	41	78.8	
J	Yes	3	37.5	17	32.7	0.999
	No	5	62.5	35	67.3	

## Data Availability

Study data will be shared if requested.

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
