# Peer review of "Investigation of Microvascular Involvement Through Nailfold Capillaroscopic Examination in Children with Familial Mediterranean Fever"

_medicina, 2025, doi:10.3390/medicina61020264_

Round 1

Reviewer 1 Report

Comments and Suggestions for Authors

Dear authors,

Your study is original and provides valuable information for the scientific community. However, it would benefit from significant language revision and clarification regarding the methodology. Please find my suggestions below.

Good luck!

The manuscript contains numerous language errors, including the title. Some other examples of language issues are: “Adherense to treatment”, “Microhemorrages”, “Neoangiogenezis”, and table 1, which contains two untranslated terms.

As a rheumatologist, I find it misleading to refer to a nailfold examination with a dermatoscope as “videocapillaroscopy” simply because the cell phone is recording in video mode. Videocapillaroscopy is a technique that relies on sophisticated tools and differs significantly from nailfold dermoscopy, as reported by Lacarrubba and colleagues (DOI: 10.1111/j.1365-4632.2010.04581.x). I believe this should be corrected.

Your study used urine dipstick tests for protein as a marker of proteinuria. While this method is valuable for daily triage in clinical practice, don’t you think it is somewhat unsuitable for research purposes?

Who performed the nailfold examination? Given that the study is neither randomized nor blinded, it seems plausible that investigation bias may have occurred. What measures were taken to minimize this issue? Please consider reporting them.

Consider displaying the p values in the first tables for easier reference.

Reconsider if Figure 5 is truly necessary.

Consider making the first paragraphs of the discussion more concise, as they address topics not directly related to the main objective of the study.

You state that nailfold capillaroscopy (NFC) was performed in accordance with previously published guidelines—please cite the guidelines utilized.

The study by Dursun and colleagues found a positive correlation between disease activity and nailfold microvascular abnormalities. On the other hand, your study enrolled primarily inactive patients and also found microvascular abnormalities. How could this apparent paradox be addressed?

The various findings of capillaroscopy are all interpreted as microcirculation issues. However, microhemorrhages and avascular areas have distinct pathogenic mechanisms and, therefore, different implications. Perhaps you could explore in greater detail the significance of each finding, linking them to the pathogenesis of FMF.

Placing the limitations after the conclusions is unfamiliar to me. Unless this was a recommendation of the journal, consider integrating them throughout the discussion.

Comments on the Quality of English Language

The manuscript contains numerous language errors, including the title. Some other examples of language issues (there are more) are: “Adherense to treatment”, “Microhemorrages”, “Neoangiogenezis”, and table 1, which contains two untranslated terms.

Author Response

Dear Reviewer,

Thank you for your interest. My answers to your comments are below. Best regards

Comment 1: The manuscript contains numerous language errors, including the title. Some other examples of language issues are: “Adherense to treatment”, “Microhemorrages”, “Neoangiogenezis”, and table 1, which contains two untranslated terms.

Answer 1: Language edited, spelling errors corrected.

Comment 2: As a rheumatologist, I find it misleading to refer to a nailfold examination with a dermatoscope as “videocapillaroscopy” simply because the cell phone is recording in video mode. Videocapillaroscopy is a technique that relies on sophisticated tools and differs significantly from nailfold dermoscopy, as reported by Lacarrubba and colleagues (DOI: 10.1111/j.1365-4632.2010.04581.x). I believe this should be corrected.

Answer 2:  The sections referring to "video-capillaroscopy" in our manuscript were revised to "nailfold capillaroscopy."

Comment 3: Your study used urine dipstick tests for protein as a marker of proteinuria. While this method is valuable for daily triage in clinical practice, don’t you think it is somewhat unsuitable for research purposes?

Answer 3:  The gold standard method for evaluating proteinuria is the quantitative measurement of protein concentration in 24-hour urine collection. However, studies have reported that collecting 24-hour urine samples is challenging and cumbersome. Instead, the protein-to-creatinine ratio in a spot urine sample has been suggested as a quantitative method for evaluating proteinuria. Additionally, the USA National Kidney Foundation (NKF) recommends using the dipstick test on spot urine samples for proteinuria screening and monitoring in both children and adults. It has also advised confirming proteinuria with quantitative methods within three months for patients with a positive dipstick test (1). In our study, proteinuria was confirmed by quantitatively measuring the protein concentration in 24-hour urine samples in all patients with a positive dipstick test or a spot urine protein/creatinine ratio greater than 0.2.

Comment 4: Who performed the nailfold examination? Given that the study is neither randomized nor blinded, it seems plausible that investigation bias may have occurred. What measures were taken to minimize this issue? Please consider reporting them.

Answer 4: The same procedure was applied to all cases before capillaroscopic evaluation. All cases were allowed to sit comfortably with their hands at heart level in a room maintained at standard temperature (20–22°C) for 15–20 minutes before capillaroscopic evaluation. Capillaroscopic evaluation of all cases was performed by the same dermatologist who was blinded to whether the cases belonged to the control or patient group. This statement was added to the Methods section of the article.

Comment 5: Consider displaying the p values in the first tables for easier reference.

Answer 5: P-values have been incorporated into Table 3 to enhance the scientific flow of the manuscript.

Comment 6: Reconsider if Figure 5 is truly necessary.

Answer 6: Figure 5 was removed.

Comment 7: Consider making the first paragraphs of the discussion more concise, as they address topics not directly related to the main objective of the study.

Answer 7: The initial paragraphs of the discussion section were made more concise.

Comment 8: You state that nailfold capillaroscopy (NFC) was performed in accordance with previously published guidelines—please cite the guidelines utilized.

Answer 8: Guidelines related to nailfold capillaroscopy are available in Reference 10. Since the subsequent sentence also cited Reference 10, a separate citation was not provided. A reference number was added to the end of the sentence discussing the guidelines in the Methods section.

Comment 9: The study by Dursun and colleagues found a positive correlation between disease activity and nailfold microvascular abnormalities. On the other hand, your study enrolled primarily inactive patients and also found microvascular abnormalities. How could this apparent paradox be addressed?

Answer 9:  In the study by Dursun et al., FMF cases in the active phase, cases in the remission phase, and the control group were compared. In our study, FMF cases in the remission phase were compared with the control group. As stated in our study, it is known that inflammation continues during remission in FMF patients. Therefore, it is anticipated that microvascular abnormalities may also persist during remission. Considering that microvascular changes may be chronic findings (2), it is likely that microvascular abnormalities detected during the active phase persist into remission. Notably, the most significant limitation of the study by Dursun et al. was the failure to reevaluate cases with microvascular abnormalities detected in the active phase during the inactive phase. If such cases were reevaluated in remission, it is plausible that microvascular changes would persist. This limitation in their study may have led to such a perception.

Comment 10: The various findings of capillaroscopy are all interpreted as microcirculation issues. However, microhemorrhages and avascular areas have distinct pathogenic mechanisms and, therefore, different implications. Perhaps you could explore in greater detail the significance of each finding, linking them to the pathogenesis of FMF.

Answer 10: Details regarding the pathogenic mechanisms of microhemorrhages and avascularization were added to the discussion section.

Comment 11: Placing the limitations after the conclusions is unfamiliar to me. Unless this was a recommendation of the journal, consider integrating them throughout the discussion.

Answer 11: Limitations were integrated into the relevant parts of the discussion section.

References

  1. Kamińska J, Dymicka-Piekarska V, Tomaszewska J, Matowicka-Karna J, Koper-Lenkiewicz OM. Diagnostic utility of protein to creatinine ratio (P/C ratio) in spot urine sample within routine clinical practice. Crit Rev Clin Lab Sci. 2020 Aug;57(5):345-364. doi: 10.1080/10408363.2020.1723487.
  2. Ma Z, Mulder DJ, Gniadecki R, Cohen Tervaert JW, Osman M. Methods of Assessing Nailfold Capillaroscopy Compared to Video Capillaroscopy in Patients with Systemic Sclerosis-A Critical Review of the Literature. Diagnostics (Basel). 2023 Jun 28;13(13):2204. doi: 10.3390/diagnostics13132204.

Reviewer 2 Report

Comments and Suggestions for Authors

This a study on capillaroscopic findings in a heterogeneous group of pediatric patients with FMF. The results show differences in the group of patients compared to the control subjects. However:

a. the two groups of patients are not similar, ie the controls and the patients differ in their nail biting habits, which in fact is not even mad clear in the table because it is not written in English

b. elevated SAA levels is written in the table; please specify numerically

c. despite response to treatment, pathological findings are more frequent; please justify

d. proteinuria is not related to microangiopathy, which conflicts current literature; please justify

e. presence or absence of dilated capillaries is not mentioned at all

The most significant caveat of the study is that it involves pediatric patients where capillaroscopy and normal vs abnormal findings is not even standardized.

Author Response

Dear reviewer,

Thank you for your interest. Your answers to your comments are below.

Best regards.

Comment 1: The two groups of patients are not similar, ie the controls and the patients differ in their nail biting habits, which in fact is not even mad clear in the table because it is not written in English.

Answer 1: When we divided all cases in our study into two groups—those with and without a nail-biting habit—no significant difference was found in capillaroscopic findings between the two groups. Furthermore, we did not find any studies in the literature demonstrating that nail-biting habits cause capillaroscopic changes.
The Turkish phrase in the table was translated into English.

Comment 2: Elevated SAA levels is written in the table; please specify numerically.

Answer 2: SAA levels were added to Table 3.

Comment 3: Despite response to treatment, pathological findings are more frequent; please justify.

Answer 3: The reason for the frequent observation of pathological findings despite treatment response was added to the final section of the discussion.

Comment 4: Proteinuria is not related to microangiopathy, which conflicts current literature; please justify.

Answer 4: In the review conducted by La Bella et al., it was reported that non-amyloid kidney disease is commonly observed in FMF patients. Conditions such as focal segmental glomerulosclerosis and IgA nephropathy seen in FMF have been reported to cause proteinuria due to vascular endothelial damage and thrombotic microangiopathies (1). This demonstrates the relationship between microangiopathy and proteinuria.

Comment 5: Presence or absence of dilated capillaries is not mentioned at all.

Answer 5: None of our cases showed capillary dilatation. This information was added to the results section.

Comment 6: The most significant caveat of the study is that it involves pediatric patients where capillaroscopy and normal vs abnormal findings is not even standardized.

Answer 6: The limitation of our study regarding the lack of standardization for normal and abnormal capillaroscopic findings in pediatric cases was added to the discussion section.

Reference

  1. La Bella, S., Di Ludovico, A., Di Donato, G., Scorrano, G., Chiarelli, F., Vivarelli, M., & Breda, L. (2023). Renal involvement in monogenic autoinflammatory diseases: A narrative review. Nephrology28(7), 363-371. doi: 10.1111/nep.14166.

Round 2

Reviewer 1 Report

Comments and Suggestions for Authors

Thank you for your attention addressing my points.

Best regards.

Reviewer 2 Report

Comments and Suggestions for Authors

Please make a remark in the manuscript regarding the causes of proteinuria in FMF patients.

Besides that, the manuscript has been adequately revised.